# Voluntary Modulation of Evoked Responses Generated by Epidural and Transcutaneous Spinal Stimulation in Humans with Spinal Cord Injury

**DOI:** 10.3390/jcm10214898

**Published:** 2021-10-24

**Authors:** Jonathan S. Calvert, Megan L. Gill, Margaux B. Linde, Daniel D. Veith, Andrew R. Thoreson, Cesar Lopez, Kendall H. Lee, Yury P. Gerasimenko, Victor R. Edgerton, Igor A. Lavrov, Kristin D. Zhao, Peter J. Grahn, Dimitry G. Sayenko

**Affiliations:** 1Mayo Clinic Graduate School of Biomedical Sciences, Mayo Clinic, Rochester, MN 55905, USA; jonathan_calvert@brown.edu; 2Department of Physical Medicine and Rehabilitation, Mayo Clinic, Rochester, MN 55905, USA; gill.megan@mayo.edu (M.L.G.); linde.margaux@mayo.edu (M.B.L.); veith.daniel@mayo.edu (D.D.V.); thoreson.andrew@mayo.edu (A.R.T.); lopez.cesar@mayo.edu (C.L.); lee.kendall@mayo.edu (K.H.L.); zhao.kristin@mayo.edu (K.D.Z.); grahn.peter@mayo.edu (P.J.G.); 3Department of Neurologic Surgery, Mayo Clinic, Rochester, MN 55905, USA; lavrov.igor@mayo.edu; 4Department of Physiology and Biomedical Engineering, Rochester, MN 55905, USA; 5Pavlov Institute of Physiology of Russian Academy of Sciences, 199034 St. Petersburg, Russia; yury.gerasimenko@louisville.edu; 6Department of Physiology and Biophysics, University of Louisville, Louisville, KY 40292, USA; 7Department of Integrative Biology and Physiology, University of California Los Angeles, Los Angeles, CA 90095, USA; vre@ucla.edu; 8Department of Neurobiology, University of California Los Angeles, Los Angeles, CA 90095, USA; 9Department of Neurosurgery, University of California Los Angeles, Los Angeles, CA 90095, USA; 10Brain Research Institute, University of California Los Angeles, Los Angeles, CA 90095, USA; 11Institut Guttmann, Hospital de Neurorehabilitació, Institut Universitari Adscrit a la Universitat Autònoma de Barcelona, 08916 Badalona, Spain; 12Centre for Neuroscience and Regenerative Medicine, Faculty of Science, University of Technology Sydney, Ultimo 2007, Australia; 13Center for Neuroregeneration, Department of Neurosurgery, Houston Methodist Research Institute, Houston, TX 77030, USA

**Keywords:** spinal cord injury, electrically evoked spinal motor potentials, spinal cord stimulation, neuromodulation

## Abstract

Transcutaneous (TSS) and epidural spinal stimulation (ESS) are electrophysiological techniques that have been used to investigate the interactions between exogenous electrical stimuli and spinal sensorimotor networks that integrate descending motor signals with afferent inputs from the periphery during motor tasks such as standing and stepping. Recently, pilot-phase clinical trials using ESS and TSS have demonstrated restoration of motor functions that were previously lost due to spinal cord injury (SCI). However, the spinal network interactions that occur in response to TSS or ESS pulses with spared descending connections across the site of SCI have yet to be characterized. Therefore, we examined the effects of delivering TSS or ESS pulses to the lumbosacral spinal cord in nine individuals with chronic SCI. During low-frequency stimulation, participants were instructed to relax or attempt maximum voluntary contraction to perform full leg flexion while supine. We observed similar lower-extremity neuromusculature activation during TSS and ESS when performed in the same participants while instructed to relax. Interestingly, when participants were instructed to attempt lower-extremity muscle contractions, both TSS- and ESS-evoked motor responses were significantly inhibited across all muscles. Participants with clinically complete SCI tested with ESS and participants with clinically incomplete SCI tested with TSS demonstrated greater ability to modulate evoked responses than participants with motor complete SCI tested with TSS, although this was not statistically significant due to a low number of subjects in each subgroup. These results suggest that descending commands combined with spinal stimulation may increase activity of inhibitory interneuronal circuitry within spinal sensorimotor networks in individuals with SCI, which may be relevant in the context of regaining functional motor outcomes.

## 1. Introduction

Transcutaneous (TSS) and epidural spinal stimulation (ESS) are electrical neuromodulation approaches that have previously been used to modulate spinal sensorimotor networks in humans [1,2]. Both TSS and ESS have been shown to enable motor functions previously thought to be permanently lost in individuals with paraplegia due to spinal cord injury (SCI), such as voluntary movement of previously paralyzed limbs [3,4,5,6,7,8], standing [9,10,11,12], and stepping [13,14,15]. TSS and ESS are both hypothesized to increase the level of excitability below the injury level, allowing previously silent, intact neural tissue that remains following injury to access sensorimotor networks responsible for function below the injury [16,17]. TSS and ESS have been shown to recruit common neural structures in electrophysiological [18] and computational modeling studies [19,20]. However, the ability of individuals with SCI to modulate epidural and transcutaneous spinally evoked motor potentials has not been investigated in detail (Figure 1A).

Previous reports of ESS and TSS have investigated spinally evoked responses via electromyography (EMG) of upper [21,22] and lower-extremity [4,23,24] musculature to characterize the effect of electrode location, different stimulation parameters, and body position on the motor thresholds and gain properties of sensorimotor networks. In these studies, stimulation was applied at low frequency ranges (0.2–2 Hz) in order to evaluate sensorimotor output while minimizing the effects of post-activation depression from frequent stimulation [25]. Previous reports indicate that some study participants, clinically diagnosed as having a motor complete SCI, were able to show signs of a non-specific, generalized increase in EMG activity below their injury level when asked to perform a full body muscle contraction by maximally flexing the muscles rostral to the SCI [7,13,26]. This has brought renewed focus to discomplete injuries, where study participants demonstrate motor activity via EMG in specific reinforcement tasks, despite being clinically classified in the ASIA (American Spinal Injury Association) Impairment Scale (AIS) as having a motor complete SCI [26,27]. Study participants without a SCI have demonstrated increased, as well as decreased, amplitude of TSS-evoked responses in some muscles during voluntary tasks [21,28,29]. However, the effect of voluntary control in individuals with SCI over TSS- or ESS-evoked responses has yet to be examined.

Here, we investigated the effect of voluntary control on TSS- and ESS-evoked responses in individuals with SCI at a range of injury severities. Participants were tested in two different conditions while supine: relaxed and while attempting maximal voluntary flexion of the lower extremities. During these tasks, spinally evoked motor potentials were recorded via EMG from the lower extremities. As previous work has demonstrated that individuals with SCI can increase the amplitude of EMG recordings taken from below the SCI, we hypothesized that voluntary attempts would increase spinally evoked response amplitude when compared to the relaxed condition.

## 2. Methods

### 2.1. Description of Participants

The experimental procedures described herein were approved by the respective University of California, Los Angeles (UCLA) and Mayo Clinic institutional review boards, and study participants provided written, informed consent to the experimental procedures. Data from two independent investigations were retrospectively analyzed via collaborative efforts from investigators at both institutions. Experiments were conducted in nine participants (seven at UCLA, two at Mayo Clinic) with chronic SCI (see Table 1 for full demographics). Study participants sustained an SCI at least two years prior to study enrollment. Two study participants were part of a study at the Mayo Clinic whose functional motor responses have previously been reported [4,7,13,30,31]. These publications focused on motor outputs during functional tasks such as voluntary control of lower-extremity muscles, stepping, standing, and sitting [7,13,30,31], as well as intraoperative recordings [4]. All data and analyses from these participants in this report were recorded at low (0.2–2 Hz) non-functional stimulation frequencies while the subjects were supine. All data contained within this manuscript have not previously been published. Briefly, these study participants performed six months of task-specific training, including body weight supported treadmill and over ground training without stimulation. At the initiation and conclusion of these six months, TSS was applied at the T10-L1 spinal vertebral levels to assess the sensorimotor connectivity of the lower-extremity musculature and spinally evoked motor responses prior to implantation of the epidural stimulator. Following these six months, participants were implanted with an epidural stimulator (Specify 5-6-5, Medtronic, Fridley, MN, USA) [4] and performed 12 months of multi-modal rehabilitation which paired task-specific rehabilitation with ESS [7]. The other seven participants were part of a study on the effects of TSS on trunk stability and self-assisted standing at the University of California, Los Angeles [12,32]. However, all data and analysis in this report are unpublished, and the study participants did not receive spinal stimulation prior to study enrollment.

### 2.2. Data Acquisition

Surface electromyogram (EMG) signals were recorded using bipolar self-adhesive electrodes placed longitudinally over the muscle belly of the vastus lateralis (VL), medial hamstrings (MH), tibialis anterior (TA), and soleus (SOL) muscles of each leg. Signals were differentially amplified and digitized at a sampling rate of 4000 samples per second (PowerLab, ADInstruments, Dunedin, New Zealand) and stored electronically (LabChart, ADInstruments, Dunedin, New Zealand). EMG data were analyzed offline using custom code written in MATLAB (Version R2020a, The Mathworks Inc., Natick, MA, USA) following application of a notch filter at 60 Hz and a 2nd order bandpass filter between 10 and 1000 Hz. All EMG recordings were synchronized to each pulse of TSS or ESS via stimulus artifact recorded from an electrode placed on the surface of the thoracolumbar spine.

Study participants were instructed to perform two experimental tasks with and without spinal stimulation: (1) to stay relaxed while lying supine to establish a control condition, and (2) to put forth maximum effort in attempting a single leg flexion maneuver including hip flexion, knee flexion, and ankle dorsiflexion simultaneously. A subset of subjects was also asked to perform joint-specific movements (e.g., plantarflexion, dorsiflexion) in the presence of stimulation. Each task was performed for at least three trials in each leg by each participant. During voluntary tasks, stimulation was delivered at a global motor threshold, which was defined as the stimulation amplitude where the peak-to-peak amplitude of all recorded muscles exceeded 20 μV responses.

### 2.3. Stimulation Procedures

Transcutaneous spinal stimulation was delivered either using a DS7A Biphasic Constant Current Stimulator (Digitimer, Hertfordshire, UK) or a custom-built, three channel constant-current stimulator. Stimulation was administered via self-adhesive electrodes (PALS, Axelgaard Manufacturing Co., Ltd., Fallbrook, CA, USA) with a diameter of 3.2 cm placed on the skin at the spinal midline between spinous processes from the T11 to L2 vertebrae to act as cathodes. Two 5 cm × 10 cm self-adhesive electrodes (PALS, Axelgaard Manufacturing Co., Ltd., Fallbrook, CA, USA) were placed symmetrically on the skin longitudinally over the abdomen for use as anodes. During TSS, stimuli were delivered as monophasic rectangular pulses with a 1 ms pulse width. Stimuli were delivered at 0–150 mA at stimulation frequencies between 0.2 and 2 Hz. A minimum of three stimuli were delivered during each trial.

Epidural spinal stimulation (ESS) was delivered using an implantable spinal cord stimulator (Specify 5-6-5, Medtronic, Fridley, MN, USA) placed between the T11-L1 vertebral bodies connected to an implanted pulse generator (RestoreSensor Sure-Scan MRI, Medtronic, Fridley, MN, USA). During ESS, stimuli were delivered as biphasic charge-balanced rectangular pulses with a 0.21 ms pulse width at a frequency of 0.2–2 Hz. Each electrode could be configured as a cathode, anode, or off. The electrode configurations were defined empirically based on the motor outputs of each subject, and were used to target specific rostral-caudal locations of the spinal cord that would enable either specific motor activation of proximal or distal lower-extremity musculature, or non-specific activation of multiple muscles of the lower extremity. ESS-evoked motor response recordings were captured during multiple ESS configurations and stimulation parameters with wide or local current distributions at the rostral and caudal ends of the electrode array (0–10 V).

### 2.4. Data Processing and Statistics

Mean and standard deviation values were calculated from at least 3 consecutive stimuli. Magnitudes of the spinally evoked potentials were calculated by measuring the area under the curve by applying a trapezoidal numerical integration to rectified EMG signals from 5 to 45 ms after the stimulus to capture the entire evoked response and prevent stimulation artifact contaminating the EMG signal. The evoked responses during voluntary contraction were normalized to the response in each muscle during the relaxed condition to account for individual differences during EMG collection in each participant. Statistically significant differences across the entire population of subjects were determined using the Wilcoxon signed-rank test for all EMG data (*p* < 0.05) using the signrank function in MATLAB, as the data were not normally distributed. The data used for the statistical tests were calculated by taking the average normalized area under the curve value of the first three evoked responses for each of the 9 subjects within the study population. After the average value was obtained for each participant, these data were entered into the signrank function to calculate the *p*-values for each recorded muscle. The paired, two-sided Wilcoxon signed-rank test was chosen over the Wilcoxon rank-sum test, as the data were from matched samples. However, comparisons across population subgroups did not have a large enough sample size to confirm statistical significance. Raw and processed datasets are available from the corresponding author upon request.

## 3. Results

### 3.1. Epidural and Transcutaneous Spinal Stimulation in the Same Participants

When stimulation was delivered at similar intensities at different electrode configurations, TSS applied at the T11/T12 intervertebral location and ESS applied at a focal, rostral portion of the electrode array (−5/+6) resulted in distinct evoked responses in the VL with relatively little activation in the other recorded muscles (MH, TA, SOL) (Figure 1B). When ESS was set with a wide field configuration (−5/+10) at the same stimulation intensity, all recorded muscles (VL, MH, TA, SOL) were activated.

### 3.2. Effect of Voluntary Effort on Spinally Evoked Responses

As shown in a representative ESS study participant and a representative TSS study participant, stimulation at motor threshold resulted in evoked responses in the leg muscles while the participants were relaxed (Figure 2A). However, when the participants were instructed to perform a full leg flexion, lower-extremity muscle responses were decreased compared to the relaxed condition. The data were normalized to compare across all participants, and the average area under the curve of the first three evoked responses was calculated for each of the nine study participants. When compared across the entire study population, the average area under the curve of the evoked responses was significantly lower across all recorded EMG muscles during the voluntary attempts to perform the leg flexion compared to the relaxed condition (mean ± standard error, *p*-value; VL: 0.6801 ± 0.1110, *p* = 0.0117; MH: 0.7084 ± 0.1157, *p* = 0.0391; TA: 0.6208 ± 0.1327, *p* = 0.0391; SOL: 0.4545 ± 0.1048, *p* = 0.0039) (Figure 2B). Furthermore, a representative subject who was asked to perform joint-specific movements (i.e., plantarflexion and dorsiflexion) demonstrated inhibition of the evoked potentials across muscles on both sides of the body during both plantarflexion and dorsiflexion (Figure 3).

### 3.3. Effect of Stimulation Modality and Injury Severity on Voluntary Modulation of Evoked Responses

To examine if stimulation modality and injury severity had an effect on the ability to modulate the evoked responses, study participants were stratified into three groups: ESS with participants diagnosed with an AIS-A SCI, TSS with AIS-A SCI, and TSS with AIS-B/C SCI. When the evoked responses were averaged across the entire voluntary contraction, both participants with AIS-A tested with ESS decreased the amplitude of their evoked responses when instructed to perform a full leg flexion (Figure 4). All participants tested with TSS were exposed to stimulation with the cathode positioned between the T12-L1 vertebral bodies. Both ESS participants used a symmetric 9+/10− configuration. In all three AIS-A participants tested with TSS, the amplitude of the evoked responses in at least 3 out of 4 of the recorded muscles did not fall outside of the standard deviation of the normalized relaxed value. However, all four AIS-B/C participants tested with TSS demonstrated a reduction in the evoked responses amplitude compared to the normalized relaxed value in at least 3 out of 4 of the recorded muscles. However, statistical comparisons across subgroups could not be made due to the low number of study participants in each subgroup.

## 4. Discussion

ESS and TSS have demonstrated improvements across a wide range of functions in individuals with SCI [3,4,6,9,11,12,13,14,15,32,33]. However, the complex interactions between stimulus pulses, the descending commands originating above the SCI and passing through the lesion site, and afferent inputs during movements to produce the functional spinal sensorimotor network outputs remain poorly understood. Here, we demonstrate the inhibition of evoked responses from ESS and TSS during voluntary attempts of individuals with severe SCI to move paralyzed limbs while lying supine.

In study participants who were stimulated with both TSS and ESS, similar evoked muscle responses were observed when the subjects were instructed to relax (Figure 1). ESS and TSS have previously been shown to activate common neural structures in electrophysiological studies [18]. Furthermore, ESS and TSS have both been shown to preferentially activate rostral-caudal and medio-lateral spinal motor pools [34,35,36,37], and both modalities are proposed to function, in part, through activation of dorsal roots entering the spinal cord [19,20,38]. However, it remains unknown what degree of specificity in activation of particular motor pools is necessary to achieve a given level of functional restoration of movement. It can be reasoned that either a specific or a broad activation pattern may be useful in engaging sensorimotor circuitry necessary for different functional tasks. Further studies are needed to demonstrate functional differences between TSS and ESS within the same individuals to effectively evaluate the advantages and disadvantages between these two modalities which may aid in choosing which strategy best fits a given individual’s injury profile and goals. Based on the currently published data, the option to choose a modality will likely result in the most desirable patient-specific outcome.

Interestingly, when study participants were asked to voluntarily contract their lower limbs while stimulation was being delivered above motor threshold, the responses were inhibited (Figure 2). Furthermore, during joint-specific contractions, subjects inhibited all the recorded muscles bilaterally (Figure 3). Previous results using TSS in individuals without an SCI have indicated inhibition of responses during passive muscle stretching and muscle-tendon vibration, and facilitation of responses during voluntary muscle contraction [21,39]. Additionally, in previous TSS studies in individuals without an SCI, agonist lower-extremity muscle EMG responses were increased and antagonistic muscle responses were decreased while attempting voluntary movement [24,28]. Within our cohort of study participants with a severe SCI, it is possible that post SCI reorganization in sensorimotor mapping has altered electrophysiological outputs resulting in simultaneous activation and reciprocal inhibition of agonist and antagonistic muscles during voluntary attempts at leg flexion and joint-specific movement [40]. Interestingly, individuals with chronic SCI typically exhibit increased excitability as evidenced by spasticity and hyperreflexia following the period of areflexia and spinal shock immediately following injury [41]. Therefore, current treatments to address spasticity include pharmacological agents that are used to reduce the excitability of the spinal cord, such as baclofen [42]. Physical treatments such as stretching, range of motion exercises, and voluntary contraction in individuals with incomplete SCI have shown improvements in spasticity, likely from enhanced activation of spinal inhibitory pathways [43]. Therefore, the present data align with the concept of increased inhibitory responses during physical tasks as well as data using TSS to attenuate spasticity in individuals with SCI, which was hypothesized to work through pre-synaptic and/or post-synaptic pathways [33]. It is noteworthy that previous results have shown bilateral facilitation of evoked responses during TSS when paired with transcranial magnetic stimulation (TMS) or galvanic vestibular stimulation (GVS), which activate the corticospinal and vestibulospinal tracts, respectively [44,45,46,47]. However, the present data suggest that stimulation of spinal cord circuitry combined with ongoing voluntary commands through remaining neural pathways crossing the lesion can inhibit spinally evoked motor responses.

Furthermore, when study participants were stratified according to the stimulation modality that was used and their injury severity as measured by their AIS classification, different patterns of evoked potential modulation emerged. AIS-A participants were able to inhibit responses across all measured muscles in ESS; however, AIS-A participants tested with TSS did not demonstrate similar results. Interestingly, participants who were classified as clinically incomplete (AIS-B/C) could inhibit the responses in at least 3 out of 4 recorded muscles (Figure 4). However, these results could not be shown to be statistically significant due to the low number of subjects in each subgroup. Previous studies have indicated that study participants with motor complete or incomplete injuries could regain voluntary motor function while using ESS [3]. Additionally, previous studies have indicated that healthy individuals [28,48] and individuals with SCI [49] could modulate TSS-evoked responses during functional tasks. However, in this study, we analyze the effect of voluntary effort on evoked response amplitude in participants with both clinically complete and incomplete SCI. These results suggest that individuals with less severe injury may be able to exert greater modulation on evoked responses recorded at motor threshold in the lower extremity. However, these findings are in a small cohort of participants and further work needs to be done to understand how remaining spinal cord fiber composition may affect lower-extremity function when paired with neuromodulation therapies. Recent mechanistic studies have suggested that the recovery of function following SCI can be attributed to propriospinal [50,51] and reorganization of cortico-reticulo-spinal tracts [52]. Additionally, motor-evoked responses and muscles activated can be modulated based on the timing that the pulse is delivered within a movement in humans and animals with SCI, which may contribute to the findings presented here as the subjects remained in the supine position continuously attempting flexion across multiple joints [49,53]. Therefore, future work should focus on the role of effort at different stages from preparation to execution of the movement and identifying the contributions of different spinal tracts to the recovery of function within the SCI population.

SCI is a heterogeneous population and results may differ depending on location and severity of injury, time since injury, and age of participant, therefore, further studies into the voluntary modulation of TSS- and ESS-evoked responses across clinical diagnoses are warranted. All of our experiments used low-frequency (0.2–2 Hz) stimulation in order to evaluate the effects of stimulation and voluntary effort without post-activation depression due to frequent stimulation. However, recent studies demonstrating return of function with spinal stimulation in individuals with severe paralysis have been at higher frequencies [3,7,13,14], and the motor output during stimulation of the spinal networks at higher or lower frequency can be dramatically different, quantitatively and qualitatively [16]. It is plausible that during higher frequencies (e.g., above 25 Hz) of spinal stimulation, the excitation predominates inhibition [54], which results in voluntary movements in the presence of spinal stimulation. Additionally, recent results using TSS have indicated that repeated exposure to stimulation may increase motoneuron output [55]. All participants within this study were not trained to perform the task, and therefore may exhibit different results when part of a long-term study. Additionally, the stimulation paradigm used within TSS for this study was composed of monophasic pulses, whereas ESS was delivered using biphasic pulses. Furthermore, the global motor threshold for this study was intentionally set at a low value of 20 µV; to observe supra-motor threshold responses of motor pools projecting to different muscles, which due to their multi-segmental origin are expected to have different thresholds. This low threshold value may have affected the ability of the subjects to modulate the responses, and further work should be performed to elucidate the effect of voluntary effort on spinally evoked responses at a range of different stimulation intensities. Lastly, the results we report were generated while study participants were positioned supine; however, body positioning influences recruitment of neural structures during spinal stimulation, and future work should evaluate the effect of voluntary intent during different body positions and tasks [23,56].

## 5. Conclusions

In the present study, we found that individuals with severe SCI could modulate EMG outputs in their lower extremity, below their level of injury in the presence of spinal stimulation. During stimulation, both TSS and ESS pulses could elicit responses in lower-extremity musculature. Importantly, with low-frequency stimulation at motor threshold, both epidural and transcutaneous spinally evoked motor responses were inhibited, when participants voluntarily attempted to activate their lower-extremity muscles. However, study participants with clinically complete SCI using ESS and participants with clinically incomplete SCI using TSS demonstrated greater ability to modulate evoked responses than participants with clinically complete SCI using TSS. These results suggest the interaction of supraspinal and spinal mechanisms even in individuals with severe SCI.

## Figures and Tables

**Figure 1 jcm-10-04898-f001:**
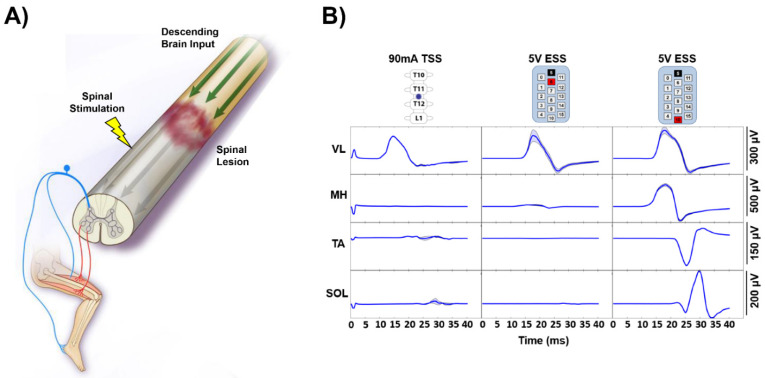
TSS- and ESS-Evoked Responses While Relaxed. (**A**). A diagram depicting inputs and outputs to the spinal cord during spinal stimulation. Descending brain input (green arrows) is interrupted by the spinal cord lesion. Spinal stimulation (yellow lightning bolt) is hypothesized to function by activating the dorsal roots carrying afferent proprioceptive information to the spinal cord. Afferent proprioceptive inputs (blue) enter the spinal cord and efferent motor outputs (red) exit the spinal cord and returns to the muscle. This figure is adapted with permission from a previous publication [6]. (**B**). While a study participant was instructed to relax while lying supine, stimulation was delivered to the same region of the spinal cord via transcutaneous spinal stimulation (TSS) and epidural spinal stimulation (ESS) using a focal and wide field. The dark line represents the average of at least three stimuli, and the shaded region indicates the ± standard deviation. VL—vastus lateralis, MH—medial hamstrings, TA—tibialis anterior, SOL—soleus, µV—microvolt, V—Volt, and mA—milliamp.

**Figure 2 jcm-10-04898-f002:**
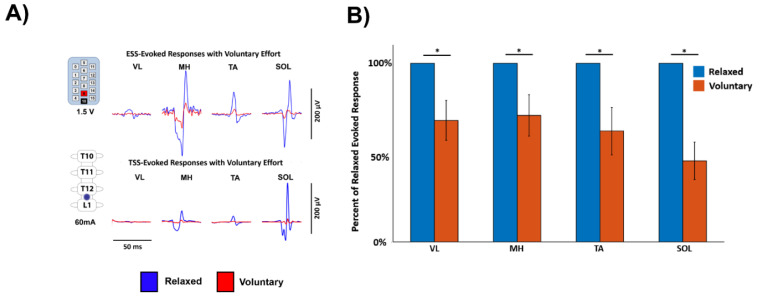
Inhibition of Evoked Response Amplitude During Voluntary Flexion. (**A**). Data from a representative study participant using ESS and a representative study participant using TSS while relaxed and while attempting maximum voluntary flexion of the lower extremities, which results in a decreased evoked response. Stimulation is delivered at the beginning of each trace. Blue indicates the relaxed condition and red indicates the voluntary flexion condition. (**B**). Grouped data from all participants within this study indicating significant decreases across all four recorded muscles when the voluntary flexion condition is compared to the relaxed condition. Data are normalized to the maximum EMG response in each muscle in each participant to compare across participants. Error bars represent the mean ± standard error. VL—vastus lateralis, MH—medial hamstrings, TA—tibialis anterior, SOL—soleus, µV—microvolt, V—Volt, mA—milliamp, and *—<0.05.

**Figure 3 jcm-10-04898-f003:**
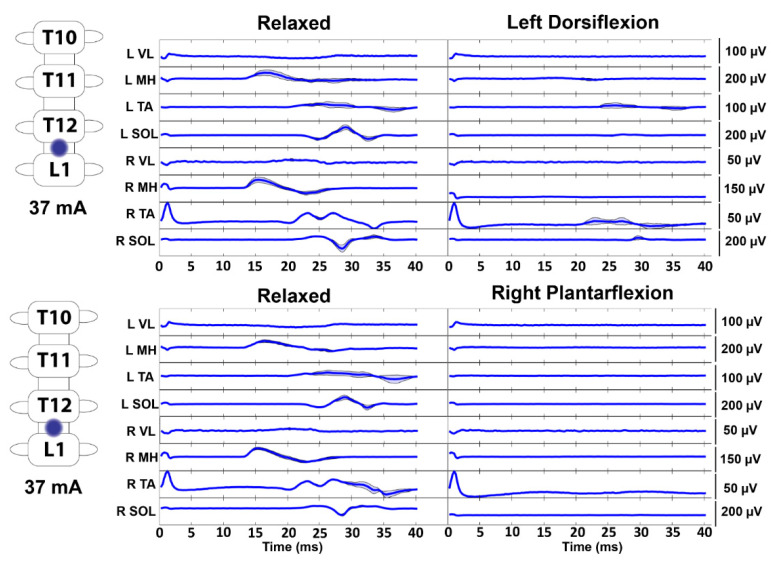
Joint-Specific Movements Decrease Motor-Evoked Responses. During T12/L1 stimulation in a representative study participant, motor-evoked responses were decreased across both the left and right lower extremities during attempts to voluntarily flex the ankle. Stimulation is delivered at the beginning of each trace. The dark line represents the average of at least three stimuli, and the shaded region indicates the ± standard deviation. VL—vastus lateralis, MH—medial hamstrings, TA—tibialis anterior, SOL—soleus, µV—microvolt, V—Volt, and mA—milliamp.

**Figure 4 jcm-10-04898-f004:**
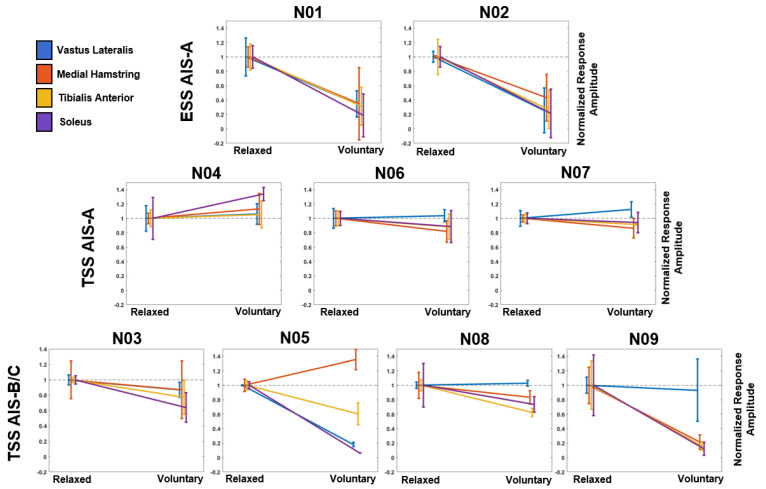
Evoked Response Modulation by ASIA Impairment Score. The first row indicates the two participants with clinically complete SCI tested with ESS. The second row indicates the three participants with clinically complete SCI tested with TSS. The third row indicates the four participants with clinically incomplete SCI tested with TSS. Data on the left of each plot refer to the average evoked response during the relaxed condition, and the data on the right refers to the average evoked response during the voluntary flexion condition. Data are normalized to the maximum EMG response in each muscle in each participant to compare across participants. The black dashed line indicates the average response of each muscle during the relaxed condition. Error bars represent the mean ± standard deviation. AIS—American Spinal Injury Association Impairment Score, VL—vastus lateralis (blue), MH—medial hamstrings (orange), TA—tibialis anterior (yellow), and SOL—soleus (purple).

**Table 1 jcm-10-04898-t001:** Study Participant Demographics.

Subject ID	Sex	Age	Injury Level	Time Since Injury	AIS Score	Stimulation Modality
N01	Male	26	T6	3 years	A	ESS, TSS
N02	Male	36	T3	6 years	A	ESS, TSS
N03	Male	22	C5	5 years	B	TSS
N04	Male	26	T2	8 years	A	TSS
N05	Female	32	C5	13 years	C	TSS
N06	Male	23	T2	4 years	A	TSS
N07	Male	25	T4	7 years	A	TSS
N08	Male	26	C4	7 years	C	TSS
N09	Male	28	T4	2 years	C	TSS

This table depicts the demographics of the study participants including their study ID, sex, age, injury level, time since injury, AIS (American Spinal Injury Association Impairment Score), and stimulation modality. ESS—epidural spinal stimulation; TSS—transcutaneous spinal stimulation.

## Data Availability

The data presented in this study are available on request from the corresponding author. The data are not publicly available due to potential patient privacy risks.

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
