# Peer review of "Voluntary Modulation of Evoked Responses Generated by Epidural and Transcutaneous Spinal Stimulation in Humans with Spinal Cord Injury"

_jcm, 2021, doi:10.3390/jcm10214898_

Round 1

Reviewer 1 Report

General comments

This is an interesting article that is well written and organized.  The figures contribute meaningfully to the interpretation of the data.  The results are particularly interesting in that reorganization that occurs after SCI seems biased toward increased, rather than decreased excitability as evidenced by the hyperreflexia/spasticity that is commonly observed. For that reason the results are counterintuitive -- it would be valuable comment on why this may be the case.  For example, the literature related to phase-dependent modulation of central pattern generated behaviors in animal models of complete spinal cord transection suggest that the timing of the stimulus pulses relative to the timing of the movement determines the motor output – could the findings be attributable to the muscles that were tested?  What are other explanations bear consideration based on what is known about the way the spinal cord reorganizes after SCI, can findings be attributed to stimulation parameters, to the activation of specific afferent circuits, etc?

Specific comments

Ln 80, here it would be valuable to point out that this study will include participants across the range of injury severity

Ln 93 It is unclear from the narrative what information has already been published related to the 2 participants at Mayo.  More complete information is needed related to the prior publication, so that the reader/editor can be alert to the possibility of redundant publication which would falsely inflate the evidence.

Ln 100, the information about why the TSS was applied is incomplete compared to the prior and subsequent information.  It is valuable to inform the reader what happened during the TSS period

152-154, wording here requires some clarification.  Does “individual electrode configurations” mean that they were participant-specific, or that pre-specified configurations were used based on prior work?

Table 1 title formatting needs attention

Ln 215  the prior section (3.2) indicated that location of stimulation influenced the output in the relaxed state,  in this section it is important for the reader to know what location of stimulation was associated with the reported responses

Ln 245 it is important to make clear that this statement refers to the relaxed state, since the same was not true during volitional effort

Ref 32 is missing information

Reviewer 2 Report

Calvert et al studied volitional influences on reflexes elicited by epidural or transcutaneous spinal stimulation in individuals with spinal cord injury. They suggest that volitional attempts to perform a whole-limb flexion results in suppression of reflexes recorded in major lower limb muscles. They further look at differences between transcutaneous and epidural stimulation and between AIS-A and AIS-B/C. 

The study is of interest, especially because it was done in humans with SCI, but there are significant methodological problems and shortcomings in the description of the methods and results that make it impossible to judge the correctness of the findings. 

I list the major issues below:

1) Statistical analysis has not been sufficiently described. There are 9 subjects and Figure 4 shows that the mean response in VL has increased in 4/9 subjects and decreased in the rest. In this case the Wilcoxon rank sum test would not be significant but a P-value of 0.0117 is reported. If several recordings from each individual were included, then this needs to be described and justified. The description of the statistics need to be very clear! Degrees of freedom and test parameters should be reported. 

2) It is reported that stimulation was applied at the global motor threshold. But Figure 1 shows example responses of TSS with a large response in VL and almost no response in other muscle. ESS is shown in a similar way but also with responses in all recorded muscles. But there is no explanation of what configuration has been used in the study. If responses are virtually absent, one cannot expect any suppression. (On the other hand, facilitation would be difficult to observe if the stimulation intensity is high; recruitment curve is saturated or close to saturation.) A threshold of 20 uV (is this an amplitude? everywhere else the area under the curve is used) seems also very low. Would it be possible that the lack of suppression in some cases is caused by the small response size?

3) Subjects N01 and N02 are shown to have received ESS and TSS but figure 4 only shows ESS data.

4) What is the rationale of grouping AIS-B/C together for the comparison in Figure 4? AIS-B and C  differ based on motor function while AIS-A and B don't.

5) It should be clearly stated in the abstract and elsewhere that the comparison between ESS and TSS and between the AIS groups are not supported with statistics (due to the low N). 

In summary, the work is potentially interesting but the analysis and presentation are severely lacking. It is impossible to judge from this text if the results are correct or not.

Round 2

Reviewer 2 Report

The authors addressed some of my comments but the main issue about the statistics has not been resolved. The Wilcoxon singed-rank test is a non-parameteric test and thus the amount of the difference has no bearing on the results, only whether the values are larger or smaller. It is still not clear how the low P-values came about. The authors also did not report the sample size and test statistic. And most importantly it is still not clear what data was used for the tests.

Author Response

Response to Reviewer #2

Comment #1: The authors addressed some of my comments but the main issue about the statistics has not been resolved. The Wilcoxon singed-rank test is a non-parameteric test and thus the amount of the difference has no bearing on the results, only whether the values are larger or smaller. It is still not clear how the low P-values came about. The authors also did not report the sample size and test statistic. And most importantly it is still not clear what data was used for the tests.

Response #1: Thank you for ensuring the accuracy and clarity of our statistical methodology. We now explicitly state in the manuscript that we use the MATLAB function “signrank” to calculate the P-values for the statistical tests (Page 4, Lines 176-181). As is recommended in the MATLAB documentation (https://www.mathworks.com/help/stats/signrank.html), as well as in the statistical literature[1], the exact Wilcoxon-signed rank method should be used when dealing with small sample sizes. The exact method does not produce a traditional test statistic such as a z-score, but calculates the p-values directly, which is why that information was not included in the manuscript. When the approximate method for calculating the Wilcoxon signed-rank test is used (recommended for large sample sizes), all of the P-values remain significant and produce an associated z-score (VL: p=0.0152, z-score=2.4286; MH: p=0.0382, z-score=2.0732; TA: p=0.0382, z-score=2.0732; SOL: p=0.0077, z-score=2.6656). However, as the sample size is small within this dataset, this information is not included in the manuscript. Furthermore, we now explicitly state within the methods and results that the data used for the statistical tests was computed based on the average normalized values of the area under the curve of the evoked responses during the relaxed and voluntary conditions for each of the 9 subjects (Page 4, Lines 176-181, Page 5, Lines 210-216).

References

  1. Harris, T.; Hardin, J.W. Exact Wilcoxon Signed-Rank and Wilcoxon Mann–Whitney Ranksum Tests. Stata J. Promot. Commun. Stat. Stata 2013, 13, 337–343, doi:10.1177/1536867X1301300208.
